# BIGRoC: Boosting Image Generation via a Robust Classifier

**Roy Ganz**                                        *ganz@campus.technion.ac.il*
*Electrical Engineering Department*
*Technion*

**Michael Elad**                                        *elad@cs.technion.ac.il*
*Computer Science Department*
*Technion*

**Reviewed on OpenReview:** *https://openreview.net/forum?id=y7RGNXhGSR*

## Abstract

The interest of the machine learning community in image synthesis has grown significantly in recent years, with the introduction of a wide range of deep generative models and means for training them. In this work, we propose a general model-agnostic technique for improving the image quality and the distribution fidelity of generated images obtained by any generative model. Our method, termed BIGRoC (Boosting Image Generation via a Robust Classifier), is based on a post-processing procedure via the guidance of a given robust classifier and without a need for additional training of the generative model. Given a synthesized image, we propose to update it through projected gradient steps over the robust classifier to refine its recognition. We demonstrate this post-processing algorithm on various image synthesis methods and show a significant quantitative and qualitative improvement on CIFAR-10 and ImageNet. Surprisingly, although BIGRoC is the first model agnostic among refinement approaches and requires much less information, it outperforms competitive methods. Specifically, BIGRoC improves the image synthesis best performing diffusion model on ImageNet $128 \times 128$ by $14.81\%$, attaining an FID score of 2.53 and on $256 \times 256$ by $7.87\%$, achieving an FID of 3.63. Moreover, we conduct an opinion survey, according to which humans significantly prefer our method's outputs.

## 1 Introduction

Deep generative models are a class of deep neural networks trained to model complicated high-dimensional data (Bond-Taylor et al., 2021). Such models receive a large number of samples that follow a certain data distribution, $x \sim P_D(x)$, and aim to produce new ones from the same statistics. One of the most fascinating generative tasks is image synthesis, which is notoriously hard due to the complexity of the natural images' manifold. Nevertheless, deep generative models for image synthesis have gained tremendous popularity in recent years, revolutionized the field, and became state-of-the-art in various tasks (Isola et al., 2017; Zhu et al., 2017; Karras et al., 2018; 2019; Brock et al., 2019; Karras et al., 2020). Energy-based models, variational autoencoders (VAEs), generative adversarial networks (GANs), autoregressive likelihood models, normalization flows, diffusion-based algorithms, and more, all aim to synthesize natural-looking images, ranging from relatively simple to highly complicated generators (Kingma & Welling, 2014; Goodfellow et al., 2014; Rezende & Mohamed, 2015; Oord et al., 2016; Ho et al., 2020).

When operating on a multiclass labeled dataset, as considered in this paper, image synthesis can be either conditional or unconditional. In the unconditional setup, the generative model aims to produce samples from the target data distribution without receiving any information regarding the target class of the synthesized images, i.e., a sample from $P_D(x)$. In contrast, in the conditional case, the generator goal is to synthesize images from a target class, i.e., a sample from $P_D(x|y)$ where $y$ is the label. As such, conditional generative models receive additional class-related information.

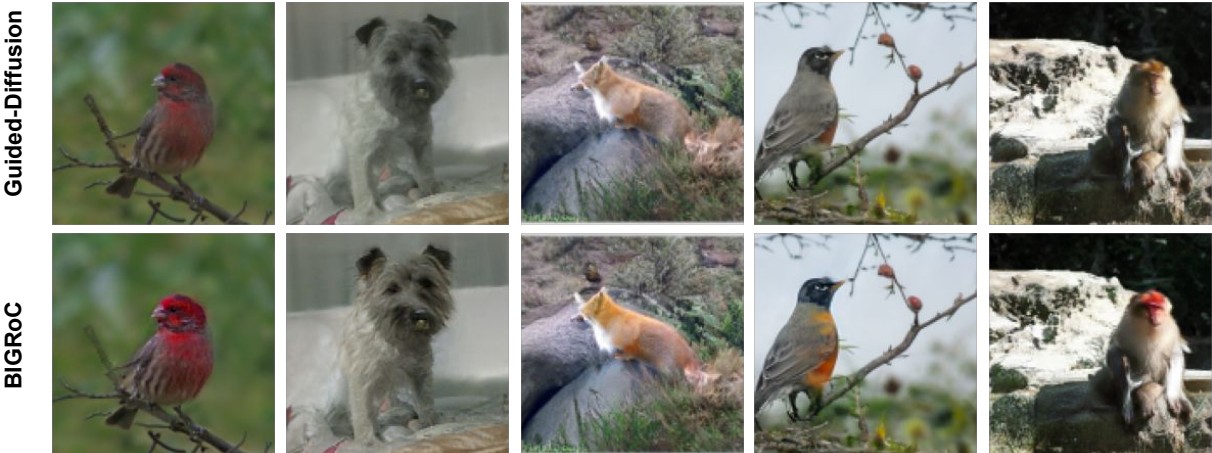

Figure 1: **Qualitative demonstration of BIGRoC**. Top: Images generated by guided diffusion trained on ImageNet $128 \times 128$. Bottom: Refined images by applying BIGRoC.

Most of the work in the deep generative models' field focused on improving the quality and the variety of the images produced by such models, tackled by seeking novel architectures and training procedures. In this work, while still aiming to improve the performance of trained generative models, we place a different emphasis than in most of these studies and propose a method for refining the outputs of such models. More specifically, our model-agnostic method improves the perceptual quality of the images synthesized by any given model via an iterative post-processing procedure driven by a *robust classifier*.

With the introduction of learning-based machines into "real-world" applications, the interest in the robustness of such models has become a central concern. While there are abundant definitions for robustness, the most common and studied is the adversarial one. This definition upholds if a classifier is robust to a small perturbation of its input, made by an adversary to fool it. Previous work (Szegedy et al., 2014; Goodfellow et al., 2015; Kurakin et al., 2017) has demonstrated that deep neural networks are not robust at all and can be easily fooled by an adversary. In light of this observation, many robustification methods were proposed, but the most popular among these is adversarial training (Goodfellow et al., 2015; Madry et al., 2018). According to this method, to train a robust classifier, one should generate adversarial examples and incorporate them into the training process. While examining the properties of such classifiers, researchers have revealed a fascinating phenomenon called *perceptually aligned gradients* (Tsipras et al., 2019). This trait implies that modifying an image to sharpen such a classifier's decision yields visual features perceptually aligned with the target class. In other words, when drifting an image content to be better classified, the changes obtained are visually pleasing and faithful to natural image content.

In this work, we harness and utilize the above-described phenomenon – we propose to iteratively modify the images created by a trained generative model so as to maximize the conditional probability of a target class approximated by a given robust classifier. This modification can potentially improve the quality of the synthesized images since it emphasizes visual features aligned with the target class, thus boosting the generation process both in terms of perceptual quality and distribution faithfulness. Due to the fundamental differences between classification and generation, we hypothesize that the robust classifier could capture different semantic features than the generative model. Thus, incorporating the two for image refinement can improve sample quality, as can be seen in Figure 1. We term this method "BIGRoC" – **B**oosting **I**mage **G**eneration via a **Ro**bust **C**lassifier.

Contrary to other refinement methods, BIGRoC is general and model-agnostic that can be applied to any image generator, both conditional or unconditional, without requiring access to its weights, given an adversarially trained classifier. The marked performance improvement achieved by our proposed method is demonstrated in a series of experiments on a wide range of image generators on CIFAR-10 and ImageNet datasets. We show that this approach enables us to significantly improve the quality of images synthesized

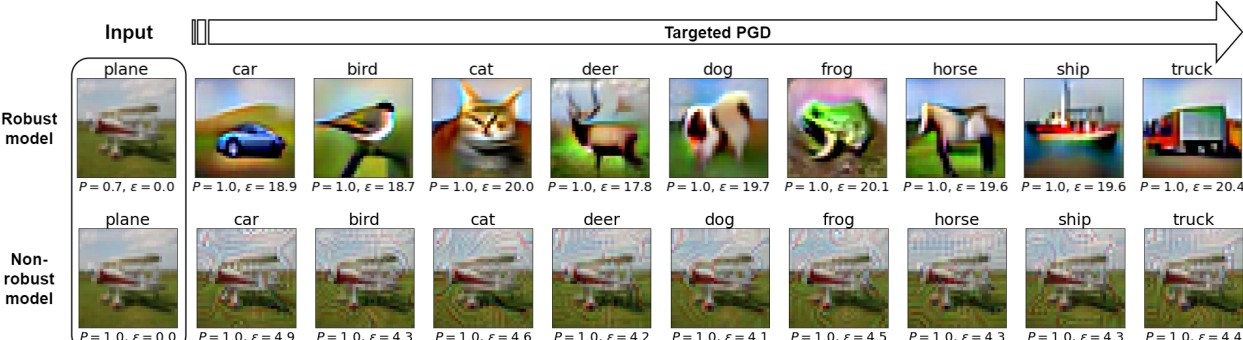

Figure 2: **PAG visualization**. Demonstration of large $l_2$-based adversarial examples on robust (top) and non robust (bottom) ResNet50 classifiers (He et al., 2016) trained on CIFAR-10. The classifier's certainty and the effective $\ell_2$ norm of the perturbation $\delta$ are denoted as P and $\epsilon$, respectively. As can be seen, robust models with PAG guides the attack towards semantically meaningful features, whereas non-robust do not.

by relatively simple models, boosting them to a level of more sophisticated and complex ones. Furthermore, we show the ability of our method to enhance the performance of generative architectures of the highest quality, both qualitatively and quantitatively. Specifically, applying BIGRoC on the outputs of guided diffusion, (Dhariwal & Nichol, 2021) significantly improves its performance on ImageNet $128 \times 128$ and $256 \times 256$ – achieving FIDs of 2.53 and 3.63, an improvement of 14.81% and 7.87%, respectively. Moreover, to truly validate that BIGRoC perceptually refines samples, we present an opinion survey, which finds that human evaluators significantly prefer the outputs of our method. As such, our work exposes the striking generative capabilities of adversarially trained classifiers. These abilities were yet to be fully discovered and utilized in previous work. To summarize, our contributions are as follow:

- We introduce BIGRoC, the first model-agnostic approach for image generation refinement by harnessing the Perceptually Aligned Gradients phenomenon.

- Extensive experiments on CIFAR-10 and ImageNet (both $128 \times 128$ and $256 \times 256$) show significant quantitative and qualitative improvements across a wide range of generative models.

- We reveal the exceptional generative power of Adversarially Robust classifiers, as the same model is capable of refining both low-quality and State-Of-The-Art image synthesizers.

## 2 Background

### 2.1 Adversarial Examples

Adversarial examples are instances intentionally designed by an attacker to cause a false prediction by a machine learning-based classifier (Szegedy et al., 2014; Goodfellow et al., 2015; Kurakin et al., 2017). The generation procedure of such examples relies on applying modifications to given training examples while restricting the allowed perturbations $\Delta$. Ideally, the "threat model" $\Delta$ should include all the possible unnoticeable perturbations to a human observer. As it is impossible to rigorously define such a set, in practice a simple subset of the ideal threat model is used, where the most common choices are the $\ell_2$ and the $\ell_\infty$ balls: $\Delta = \{\delta \; : \; \|\delta\|_{2/\infty} \leq \epsilon\}$. Given $\Delta$, the attacker receives an instance $x$ and generates $\hat{x} = x + \delta$ $s.t.$ $\delta \in \Delta$, while aiming to fool the classifier. Adversarial attacks can be both untargeted or targeted: An untargeted attack perturbs the input in a way that minimizes $p(y|\hat{x})$ with respect to $\delta$. In contrast, a targeted attack receives in addition the target class $\hat{y}$, and perturbs $x$ to maximize $p_(\hat{y}|\hat{x})$. There are diverse techniques for generating adversarial examples, yet, in this work, we focus on targeted attacks using the Projected Gradient Descent (PGD) (Madry et al., 2018)– an iterative method for creating adversarial examples that operates as shown in Algorithm 1. The operation $\Pi_\epsilon$ is the projection operator onto $\Delta$, and $\ell(\cdot)$ is the classification loss.

---

**Algorithm 1** Targeted Projected Gradient Descent (PGD)

---

**Input**: classifier $f_\theta$, input $x$, target class $\hat{y}$, $\epsilon$, step size $\alpha$, number of iterations $T$

$\delta_0 \leftarrow 0$

**for** *t from 0 to T* **do**

$\quad | \quad \delta_{t+1} = \Pi_\epsilon(\delta_t - \alpha\nabla_\delta\ell(f_\theta(x + \delta_t), \hat{y}));$

**end**

$x_{adv} = x + \delta_T$

**Output**: $x_{adv}$

---

## 2.2 Adversarial Robustness

Adversarial robustness is a property of classifiers, according to which applying a small perturbation on a classifier's input in order to fool it does not affect its prediction (Goodfellow et al., 2015). To attain such classifiers, one should solve the following optimization problem:

$$\min_\theta \sum_{x,y \in D} \max_{\delta \in \Delta} \ell(f_\theta(x + \delta), y) \tag{1}$$

Namely, train the classifier to accurately predict the class labels of the "toughest" perturbed images, allowed by the threat model $\Delta$. In practice, solving this optimization problem is challenging, and there are several ways to attain an approximated solution. The most simple yet effective method is based on approximating the solution of the inner-maximization via adversarial attacks, such as PGD (Madry et al., 2018). According to this strategy, the above optimization is performed iteratively, fixing the classifier's parameters $\theta$ and optimizing the perturbation $\delta$ for each example via PGD, and then fixing these and updating $\theta$. Repeating these steps results in a robust classifier, as we use in this work.

## 2.3 Perceptually Aligned Gradients

Perceptually aligned gradients (PAG) (Tsipras et al., 2019; Kaur et al., 2019; Aggarwal et al., 2020; Ganz et al., 2022) is a trait of adversarially trained models, best demonstrated when modifying an image to maximize the probability assigned to a target class. (Tsipras et al., 2019) show that performing the above process on such models yields meaningful features aligned with the target class. It is important to note that this phenomenon does not occur in non-robust models. The perceptually aligned gradients property indicates that the features learned by robust models are more aligned with human perception. Figure 2 presents a visual demonstration of this fascinating phenomenon.

## 3 Boosting Image Generation via a Robust Classifier

We propose a method for improving the quality of images synthesized by trained generative models, named BIGRoC: Boosting Image Generation via a Robust Classifier. Our method is model agnostic and does not require additional training or fine-tuning of the generative model that can be viewed as a post-processing step performed on the synthesized images. Thus, BIGRoC can be easily applied to any generative model, both conditional or unconditional. This mechanism harnesses the perceptually aligned gradients phenomenon to modify the generated images to improve their visual quality. To do so, we perform an iterative process of modifying the generated image $x$ to maximize the posterior probability of a given target class $\hat{y}$, $p_\theta(\hat{y}|x)$, where $p_\theta$ is modeled by an adversarially trained classifier. This can be achieved by performing a PGD-like process that instead of adversarially changing an image $x$ of class $y$ to a different class $\hat{y} \neq y$, modifies it in a way that maximizes the probability that $x$ belongs to class $y$. Therefore, our method requires a trained robust classifier that operates on the same data source as the images we aim to improve.

In the conditional generation process, the generator $G$ receives the class label $y$, from which it suppose to draw samples. Hence, in this setup, we have information regarding the class affiliation of the image, and we can maximize the corresponding conditional probability. In the unconditional generation process, the generator does not receive class labels at all, and its goal is to draw samples from $p(x)$. Thus, in this case,

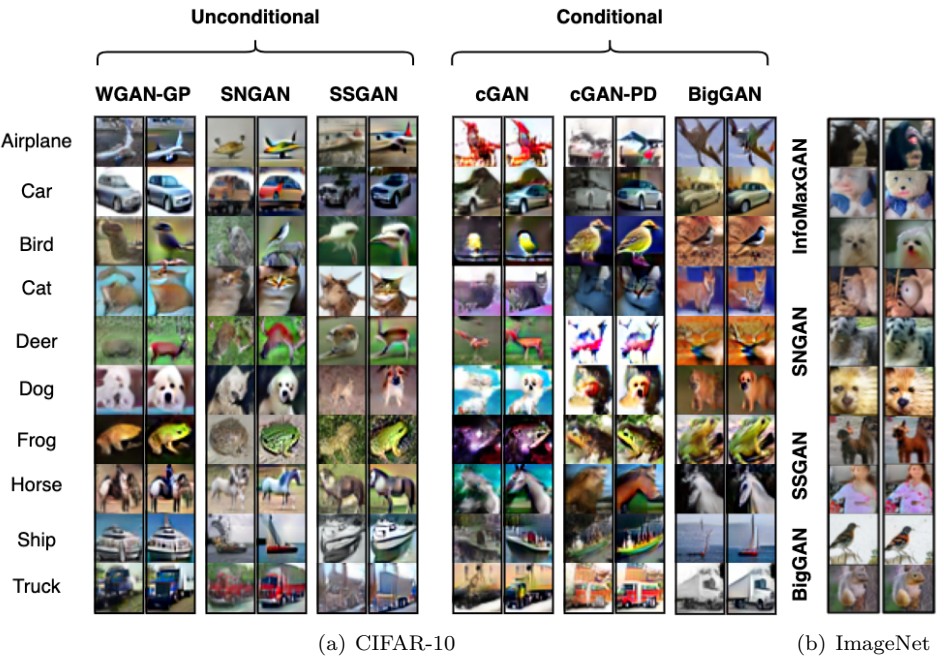

(a) CIFAR-10  (b) ImageNet

Figure 3: **BIGRoC qualitative results**. We present pairs of images where the left columns contain generated images and the right ones the boosted results. Figures 3(a) and 3(b) show such results on CIFAR-10 and ImageNet $128 \times 128$, respectively.

we cannot directly maximize the desired posterior probability, as our method suggests. To bridge this gap, we propose to estimate the most likely class via our robust classifier $f_\theta$ and afterward modify the image via the suggested method to maximize its probability. The proposed image generation boosting is described in Algorithm 2 for both setups.

While the above-described approach for unconditional sampling works well, it could be further improved. We have noticed that in this case, estimating the target classes $Y$ of $X_{gen}$ via $f_\theta$ might lead to unbalanced labels. For example, in the CIFAR-10, when generating 50,000 samples, we expect approximately 5,000 images per each of the ten classes, and yet the labels' estimation does not distribute uniformly at all. This imbalance stems from the incompetence of the generative model to capture the data distribution, which leads to a bias in the target labels estimation of the boosting algorithm, affecting the visual content of $X_{boost}$. Since quantitative evaluation is impaired by such class imbalances, this bias limits the quantitative improvement attained by BIGRoC. We emphasize that this issue is manifested only in the quantitative metrics, and when qualitatively evaluating the boosted images, the improvement is significant, as can be seen in the supplementary material.

To further enhance the quantitative results of our algorithm in the unconditional case, we propose to de-bias the target class estimation of $X_{gen}$ and attain close to uniform class estimations. A naive solution to this can be achieved by generating more samples and extracting a subset of these images with a labels-balance. This approach is computationally heavy and does not use the generated images as-is, raising questions regarding the fairness of the quantitative comparison. Thus, we propose a different debiasing technique – we modify the classifier's class estimation to become more balanced by calibrating its logits. More specifically, we shift the classifier's logits by adding a per-class pre-calculated value, $d_{c_i}$, that induces equality of the mean logits value across all classes. We define $\mathbf{d_c}$ as a vector containing all $d_{c_i}$ values: $\mathbf{d_c} = [d_{c_0}, \dots d_{c_{N-1}}]$ where $N$ is the number of classes. For simplicity, we denote $logit_{c_i}$ as the logit of class $c_i$ corresponding to a generated sample $x_{gen}$. We approximate $\mathbb{E}_{x_{gen}}[logit_{c_i}]$ for each class $c_i$, using a validation set of generated images, and calculate a per-class debiasing factor: $d_{c_i} = a - \hat{\mathbb{E}}_{x_{gen}}[logit_{c_i}]$ (WLOG, $a = 1$), where $\hat{\mathbb{E}}_{x_{gen}}[logit_{c_i}]$ is a mean estimator. After calculating $d_{c_i}$, given a generated image $x_{gen}$, we calculate its logits and add $d_{c_i}$ to

Figure 4: **Threat-model effect demonstration**. Generated images via a conditional GAN (left) perturbed via a targeted PGD to maximize the probability of their target classes (planes, cars and birds) using either $\ell_\infty$ or $\ell_2$ threat models. While the boosted outputs attained by the $\ell_\infty$ entirely change the structure of the images and lead to unnatural results, the $\ell_2$ threat model leads to better results.

it to obtain debiased logits ($\hat{logit}_{c_i}$), from which we derive the unbiased class estimation via softmax. The following equation shows that, given a correct estimation of the per-class logits' mean, the per-class means of the debiased logits are equal:

$$\mathbb{E}_{x_{gen}}[\hat{logit}_{c_i}] = \mathbb{E}_{x_{gen}}[d_{c_i} + logit_{c_i}] = \mathbb{E}_{x_{gen}}[a - \hat{\mathbb{E}}_{x_{gen}}[logit_{c_i}] + logit_{c_i}] =$$
$$a - \hat{\mathbb{E}}_{x_{gen}}[logit_{c_i}] + \mathbb{E}_{x_{gen}}[logit_{c_i}] \approx a$$

We present in Appendix F an empirical demonstration that verifies the validity of this method and shows its qualitative effects on unconditional generation boosting.

---

**Algorithm 2** BIGRoC

**Input**: Robust classifier $f_\theta$, $x_{gen}$, $y_{gen}$, $\epsilon$, step size $\alpha$, number of iterations $T$, debiasing vector $\mathbf{d_c}$

**if** $y_{gen}$ *is None* **then**

$\quad$ $y_{gen} = argmax(f_\theta(x_{gen}) + \mathbf{d_c})$

**end**

$x_{boost} = Targeted\ PGD(f_\theta, x_{gen}, y_{gen}, \epsilon, \alpha, T)$

**Output**: $x_{boost}$

---

As can be seen in Algorithm 2, it receives as input the generated images and their designated labels (if exist) and returns an improved version of them. As such, this method can be applied at the inference phase of generative models to enhance their performance in a complete separation from their training. Furthermore, BIGRoC does not require access to the generative models at all and thus can be applied to standalone images, regardless of their origin. As can be seen from the algorithm's description, it has several hyperparameters that determine the modification process of the image: $\epsilon$ sets the maximal size of the perturbation allowed by the threat model $\Delta$, $\alpha$ controls the step size at each update step, and $T$ is the number of updates. Another choice is the norm used to define the threat model $\Delta$.

The hyperparameter $\epsilon$ is central in our scheme - when $\epsilon$ is too large, the method overrides the input and modifies the original content in an unrecognizable way, as can be seen in Figure 2. On the other hand, when $\epsilon$ is too small, the boosted images remain very similar to the input ones, leading to a minor enhancement. As our goal is to obtain a significant enhancement to the synthesized images, a careful choice of $\epsilon$ should be practiced, which restricts the allowed perturbations in the threat model.

Another important choice is the threat model $\Delta$ itself. Two of the most common choices of $\Delta$ for adversarial attacks are the $\ell_\infty$ and the $\ell_2$ balls. Due to the desired behavior of our method, using the $\ell_\infty$ ball is less preferable: it allows a change of $\pm\epsilon$ to every pixel, and as such, it will not focus on meaningful specific locations, and might not preserve the existing structure of the synthesized input image. Thus, we choose the $\ell_2$ ball as our threat model, with relatively small $\epsilon$. Such a choice restricts the allowed perturbations and leads to changes that may concentrate on specific locations while preserving most of the existing content in the generated images. A visual demonstration of these considerations is given in Figure 4.

# 4 Related Work

## 4.1 Image Generation Refinement

There are two main lines of work that aim to improve the quality of generated images. One is based on rejection sampling – improving the generation quality of GANs by discarding low-quality images identified by the GAN's discriminator (Turner et al., 2019; Azadi et al., 2019). Contrary to such work that does not enhance the generated images but rather acts as a selector, BIGRoC does not discard any synthesized images and improves their quality by modifying them.

The second line of work (Tanaka, 2019; Che et al., 2021; Ansari et al., 2021), which is closely related to ours, addresses the task of sample refinement – modifying the generated images to attain improved perceptual quality. These papers propose methods for improving synthesized images using the guidance of the GAN's discriminator. More precisely, given a latent code of a generated image, their strategy is to modify it to maximize the score given by the GAN's discriminator. Therefore, to enhance the perceptual quality of a set of generated images, these approaches require access to both the generator and the discriminator weights and the corresponding latent code of the generated images. Thus, such methods rely on the existence of a discriminator, making them not model-agnostic. Moreover, as image refinement is an applicative task, this constraint prevents such methods from operating directly on images without additional information, making their configuration less realistic. In contrast to these, our work offers a much simpler and different way of boosting generated images by an external pretrained robust classifier in a completely model-agnostic way. Our method can operate without requiring access to the latent vectors generating the images or the weights of the generative model that produces them. Thus, BIGRoC can be applied to standalone images – a realistic setup where none of the existing methods can operate. Moreover, while competitive methods couple the generator with its discriminator, we show that a single robust classifier is capable of improving a wide variety of sample qualities from the same dataset, making our configuration even more realistically appealing, as it requires only one model per dataset. In addition, while (Tanaka, 2019; Che et al., 2021) are limited to GANs only, our method is model-agnostic and capable of improving generated images of any source, e.g., diffusion models. In Section 5.4, we empirically demonstrate that although the existing methods have much stricter requirements than ours, it leads to improved performance. Moreover, our approach is the first to be successfully applied to ImageNet $256 \times 256$, proving its scalability.

## 4.2 Perceptually Aligned Gradients in Computer Vision

PAG phenomenon was previously utilized for solving various computer vision tasks, such as inpainting, image translation, super-resolution, and image generation (Santurkar et al., 2019). However, in image synthesis, the obtained performance is relatively disappointing, *i.e.*, Inception Score (IS) of 7.5 on CIFAR-10, far from state-of-the-art (SOTA) performance. Moreover, the qualitative results demonstrated are far from pleasing. Thus, it raises the question if this is the generative capabilities performance limit of robust image classifiers. In our work, we provide a definite answer to this question by harnessing PAG to a different task – sample refinement. To this end, we build upon any existing generative model, including high-performing ones, and empirically show that a robust classifier can boost the performance of SOTA image generators. In particular, we demonstrate that robust classifier guidance can improve Diff BigGAN (Zhao et al., 2020) from IS of 9.48 to 9.61, well beyond the 7.5 obtained in (Santurkar et al., 2019). As such, our work exposes a much stronger force that does exist in adversarially robust classifiers in capturing high perceptual quality features.

# 5 Experimental Results

In this section, we present experiments that demonstrate the effectiveness of our method on the most common datasets for image synthesis – CIFAR-10 (Krizhevsky, 2012) and ImageNet (Deng et al., 2009). Given a generative model, we use it to synthesize a set of images $X_{gen}$ and apply our method to generate $X_{boost}$, according to Algorithm 2. Since BIGRoC is model-agnostic, it can be easily applied to any generative model, given a robust classifier trained on the same data source. We utilize the model-agnostic property to examine the effects of applying the proposed boosting over a wide variety of image generators of different

qualities: from relatively simple to sophisticated and complex ones. We test our method on both conditional and unconditional generative models to validate that the proposed scheme can enhance different synthesis procedures. An application of our method without the ground truth labels is termed BIGRoC$_{PL}$, as it generates pseudo labels (PL) using our robust classifier. In contrast, BIGRoC$_{GT}$ refers to the case where ground truth (GT) labels are available (in the conditional image synthesis).

In all the experiments, for each dataset, we use the same adversarial robust classifier to refine all the generated images of different sources. The only needed adjustment is at tuning $\epsilon$, which defines the allowed size of the visual modifications done by BIGRoC. The striking fact that a single robust classifier improves both low and high-quality images strongly demonstrates the versatility of our approach and the surprising refinement capabilities possessed by such a model. We analyze BIGRoC performance both quantitatively and qualitatively, using Fréchet Inception Distance (FID, (Heusel et al., 2017), lower is better), and Inception Score (IS, (Salimans et al., 2016), higher is better). Moreover, we conduct an opinion survey to validate that human observers find BIGRoCs outputs more pleasing. In addition, we compare our approach with other image refinement SOTA methods mentioned in Section 4.

### 5.1 CIFAR-10

In this section, we evaluate the performance of the proposed BIGRoC on the CIFAR-10 dataset, using a single publicly-available adversarially trained ResNet-50 on CIFAR-10 as the robust classifier (Engstrom et al., 2019). We experiment with a wide range of generative models, both conditional and unconditional.

**Quantitative Results** Table 1 contains BIGRoC$_{PL}$ and BIGRoC$_{GT}$ quantitative results for both unconditional and conditional image synthesis refinement, respectively. These results indicate that BIGRoC achieves a substantial improvement across a wide range of generator architectures, both conditional and unconditional, demonstrating the method's versatility and validity. Interestingly, the fact that the same robust classifier can enhance both low-quality models (that require focus on high-level features) and top-performing models (that require emphasis on low-level features) strongly attests to its generative powers.

**Qualitative Results** To strengthen the quantitative results, we show in Figure 3(a) and in the supplementary material qualitative results that verify that the "boosted" results indeed look better perceptually.

### 5.2 ImageNet

We turn to evaluate the performance of the proposed BIGRoC on the ImageNet $128 \times 128$ and $256 \times 256$ datasets, using a single publicly available adversarially trained ResNet-50 on ImageNet as the robust classifier (Engstrom et al., 2019) for both of the resolutions.

**Quantitative Results** Table 2 summarize our quantitative results on ImageNet $128 \times 128$ and $256 \times 256$. These results strongly indicate that BIGRoC is also highly beneficial for higher-resolution images from richer datasets. Specifically, BIGRoC improves the best-performing diffusion model (Dhariwal & Nichol, 2021) on ImageNet $128 \times 128$ by 14.81%, leading to FID of 2.53, and on $256 \times 256$ by 7.87%, leading to FID of 3.63. The obtained results show a similarly marked improvement in IS. Interestingly, our approach (BIGRoC$_{PL}$) is capable of refining the outputs of top-performing generative models even with the absence of the ground truth labels.

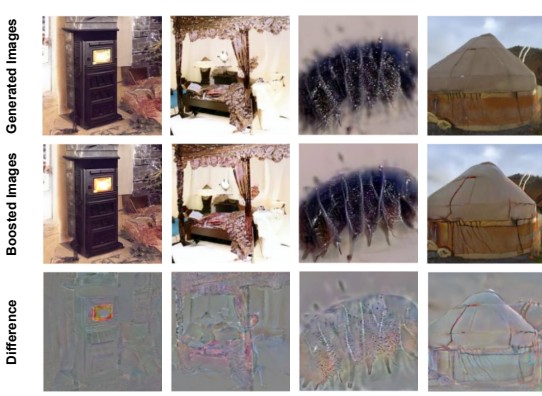

Figure 5: **Qualitative comparison on ImageNet $256 \times 256$**. Top: Images generated by guided diffusion. Middle: Images boosted by BIGRoC. Bottom: contrast stretched differences.

**Qualitative Results** To verify that our method indeed leads to better perceptual quality, we show visual results on ImageNet $128 \times 128$ in Figure 3(b) and in Figure 1. Moreover, we present in Figure 5 a qualitative comparison between images generated by a guided-diffusion trained on ImageNet $256 \times 256$ and the outputs of BIGRoC applied to them. We also show the images' differences (after contrast stretching) to better grasp

Table 1: BIGRoC quantitative results on CIFAR-10 for both conditional and unconditional generators.

| Architecture | Cond. | FID ↓ | IS ↑ |
|---|---|---|---|
| VAE (Kingma & Welling, 2014) | ✗ | $152.04 \pm 0.19$ | $3.05 \pm 0.01$ |
| w/ BIGRoC$_{\text{PL}}$ | | $88.68 \pm 0.37$ | $6.27 \pm 0.04$ |
| $\Delta$ | | **-63.36** | **+3.22** |
| DCGAN (Radford et al., 2015) | ✗ | $38.34 \pm 0.11$ | $6.10 \pm 0.01$ |
| w/ BIGRoC$_{\text{PL}}$ | | $29.93 \pm 0.05$ | $7.28 \pm 0.04$ |
| $\Delta$ | | **-8.41** | **+1.18** |
| WGAN-GP (Salimans et al., 2016) | ✗ | $22.62 \pm 0.09$ | $7.49 \pm 0.03$ |
| w/ BIGRoC$_{\text{PL}}$ | | $16.28 \pm 0.08$ | $8.15 \pm 0.03$ |
| $\Delta$ | | **-6.32** | **+0.66** |
| SNGAN (Miyato et al., 2018) | ✗ | $17.19 \pm 0.07$ | $8.04 \pm 0.02$ |
| w/ BIGRoC$_{\text{PL}}$ | | $13.25 \pm 0.10$ | $8.61 \pm 0.04$ |
| $\Delta$ | | **-3.94** | **+0.57** |
| InfoMaxGAN (Lee et al., 2021) | ✗ | $15.41 \pm 0.12$ | $8.09 \pm 0.05$ |
| w/ BIGRoC$_{\text{PL}}$ | | $11.27 \pm 0.11$ | $8.48 \pm 0.03$ |
| $\Delta$ | | **-4.14** | **+0.39** |
| SSGAN (Chen et al., 2019) | ✗ | $15.05 \pm 0.06$ | $8.20 \pm 0.02$ |
| w/ BIGRoC$_{\text{PL}}$ | | $10.77 \pm 0.06$ | $8.61 \pm 0.03$ |
| $\Delta$ | | **-4.28** | **+0.41** |
| cGAN (Mirza & Osindero, 2014) | ✓ | $29.26 \pm 0.10$ | $6.95 \pm 0.03$ |
| w/ BIGRoC$_{\text{GT}}$ | | $8.89 \pm 0.05$ | $8.57 \pm 0.05$ |
| $\Delta$ | | **-20.37** | **+1.62** |
| cGAN-PD (Mirza & Osindero, 2014) | ✓ | $11.10 \pm 0.07$ | $8.54 \pm 0.03$ |
| w/ BIGRoC$_{\text{GT}}$ | | $8.33 \pm 0.06$ | $8.76 \pm 0.05$ |
| $\Delta$ | | **-2.77** | **+0.22** |
| BigGAN (Brock et al., 2019) | ✓ | $7.45 \pm 0.08$ | $9.38 \pm 0.05$ |
| w/ BIGRoC$_{\text{GT}}$ | | $6.79 \pm 0.02$ | $9.47 \pm 0.02$ |
| $\Delta$ | | **-0.66** | **+0.09** |
| Diff BigGAN (Zhao et al., 2020) | ✓ | $4.37 \pm 0.03$ | $9.48 \pm 0.03$ |
| w/ BIGRoC$_{\text{GT}}$ | | $3.95 \pm 0.02$ | $9.61 \pm 0.03$ |
| $\Delta$ | | **-0.42** | **+0.13** |

the perceptual modifications applied by our method. As can be seen, BIGRoC focuses on the edges and textures and leads to sharper and more high-contrast images, which are more pleasing to a human observer.

## 5.3 Opinion survey with human evaluators

As current quantitative metrics are limited proxies for human perception, we conduct an opinion survey with human evaluators to better validate that our proposed method leads to better visual quality. We randomly sampled 100 images from a guided diffusion, trained on ImageNet $128 \times 128$, and applied BIGRoC on them. Human evaluators were shown pairs of such images in a random order, and the question: *"Which image looks better?"*. They were asked to choose an option between "Left", "Right", and "Same". The image pairs were in a random order for each evaluator. We collected 1038 individual answers in total, where 56.07% of the votes opted for BIGRoC's outputs, 19.17% the generated ones, and the rest chose the "same" option. When aggregating the votes per each image pair, in 83% of the pairs, human evaluators were in favor of our algorithm's outputs, while only in 12% of them, they prefer the generated ones. In the remaining 5%, the "same" option obtained the most votes. These results indicate that human evaluators considerably prefer our method outputs, with a significance level of $> 95\%$. We provide additional details regarding the conducted survey in Appendix B.

Table 2: Quantitative results on ImageNet $128 \times 128$.

| Architecture | Resolution | FID ↓ | IS ↑ |
|---|---|---|---|
| SNGAN (Miyato et al., 2018) | $128 \times 128$ | 62.28 | 13.05 |
| w/ BIGRoC$_{PL}$ | | 40.40 | 71.67 |
| Δ | | **-21.88** | **+58.62** |
| SSGAN (Chen et al., 2019) | $128 \times 128$ | 63.60 | 13.75 |
| w/ BIGRoC$_{PL}$ | | 38.93 | 73.94 |
| Δ | | **-24.67** | **+60.19** |
| InfoMaxGAN (Lee et al., 2021) | $128 \times 128$ | 60.61 | 13.79 |
| w/ BIGRoC$_{PL}$ | | 37.70 | 75.49 |
| Δ | | **-22.91** | **+61.7** |
| BigGAN-deep (Brock et al., 2019) | $128 \times 128$ | 6.02 | 145.83 |
| w/ BIGRoC$_{PL}$ | | 5.69 | 176.42 |
| w/ BIGRoC$_{GT}$ | | 5.71 | 226.17 |
| Δ | | **-0.31** | **+80.34** |
| Guided Diffusion (Dhariwal & Nichol, 2021) | $128 \times 128$ | 2.97 | 141.37 |
| w/ BIGRoC$_{PL}$ | | 2.77 | 150.43 |
| w/ BIGRoC$_{GT}$ | | 2.53 | 169.73 |
| Δ | | **-0.44** | **+28.36** |
| BigGAN-deep (Brock et al., 2019) | $256 \times 256$ | 7.03 | 202.65 |
| w/ BIGRoC$_{PL}$ | | 6.93 | 221.78 |
| w/ BIGRoC$_{GT}$ | | 6.84 | 228.23 |
| Δ | | **-0.19** | **+25.58** |
| Guided Diffusion (Dhariwal & Nichol, 2021) | $256 \times 256$ | 3.94 | 215.84 |
| w/ BIGRoC$_{PL}$ | | 3.69 | 249.91 |
| w/ BIGRoC$_{GT}$ | | 3.63 | 260.02 |
| Δ | | **-0.31** | **+44.18** |

### 5.4 Comparison with other methods

Previous works, such as DOT (Tanaka, 2019), DDLS (Che et al., 2021) and DG-$f$low (Ansari et al., 2021), address image generation refinement, as we do. As mentioned in Section 4, we propose a much simpler approach than our competitors and require much less information since BIGRoC can operate without access to the image generator, the discriminator, and without knowing the latent codes corresponding with the generated images. Moreover, BIGRoC is the first model-agnostic refinement method that can operate on any generative model's output. Besides these advantages, we demonstrate that although our method utilizes less information than other methods and can operate in setups that competitive approaches can not, BIGRoC performs on par and even better. To this end, we compare our method with current SOTA image refinement methods on CIFAR-10 and ImageNet $128 \times 128$. For both these setups, we adopt the same publicly available pretrained models of SN-ResNet-GAN as used in (Tanaka, 2019; Che et al., 2021; Ansari et al., 2021) and apply our algorithm to the generated images and evaluate its performance quantitatively (see the supplementary material for additional implementation details). In Table 3, we compare the quantitative results of the DOT, DDLS, and DG-$f$low with BIGRoC using IS, as this is the common metric reported in these papers. Our findings show that our approach significantly outperforms the other methods on the challenging ImageNet dataset while obtaining comparable results to DG-$f$low on CIFAR-10.

## 6 Ablation Study

We conduct experiments to better understand the performance improvements obtained and analyze the effects of the central hyperparameters in our algorithm.

Table 3: Quantitative comparison with competitive methods.

| Architecture | Inception Score ↑ | |
| | CIFAR-10 | ImageNet |
| --- | --- | --- |
| SN-ResNet-GAN | 8.38 | 36.8 |
| w/ DOT | – | 37.29 |
| w/ DDLS | 9.09 | 40.2 |
| w/ DG-flow | **9.35** | – |
| w/ BIGRoC | 9.33 | **44.68** |

## 6.1 The effect of $\epsilon$

As stated in Section 3, the value of $\epsilon$ defines the allowed perturbation. In Figure 6, we demonstrate the effect of different values of $\epsilon$ when applying BIGRoC$_{\mathrm{GT}}$ to images generated by guided diffusion, trained on ImageNet $256 \times 256$. As can be seen, it affects the trade-off between diversity and fidelity, demonstrated by IS versus FID values. Our method attains lower FID scores in a range of tested values of $\epsilon$ while achieving better IS, leading to better trade-offs. We present additional ablations in the supplementary materials. This study shows that although its importance, BIGRoC is not highly sensitive to the choice of $\epsilon$.

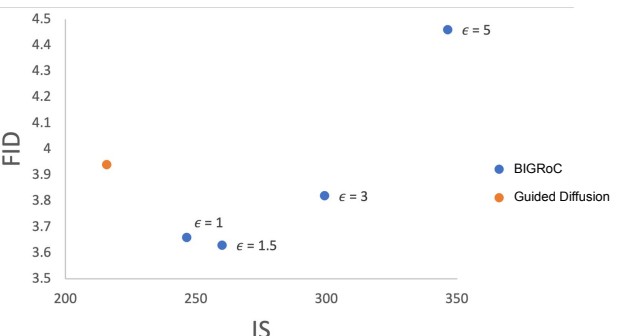

Figure 6: **Trade-off effects of $\epsilon$**. Demonstrated on Guided-Diffusion trained on ImageNet $256 \times 256$.

## 6.2 The effect of the guiding classifier

In this work, we propose a method for perceptually improving generated images by using a robust classifier. However, there are a few other choices of models that can be used for guiding the refinement process. The simplest choice is using a standard non-robust classifier trained on the same dataset as the generative model. Another applicable option when refining the outputs of guided diffusion models is to use the inherent guiding classifier of the diffusion. To validate that indeed robust classifiers have the best refining capabilities, we apply Algorithm 2 on 50,000 ImageNet $128 \times 128$ images generated by a guided diffusion (Dhariwal & Nichol, 2021), using a ResNet50 ("Vanilla clf"), a guiding classifier ("Guiding clf") proposed in (Dhariwal & Nichol, 2021) and a robust classifier ("RoC"). We summarize our positive findings in Table 4, which show that a robust classifier provides the best guidance among the tested alternatives.

Table 4: FID scores of Algorithm 2 using different classifiers.

| w/o Alg. 2 | w/ Alg. 2 | | |
| | Vanilla clf | Guiding clf | RoC |
| --- | --- | --- | --- |
| 2.97 | 3.19 | 2.77 | 2.53 |

## 6.3 The effect of the threat model

In all of our experiments, we use an adversarially trained (AT) robust classifier with a threat model $\Delta$ based on $\ell_2$ norm with a predefined $\epsilon$ value. In this section, we study the effect of $\epsilon$ in the training of the AT robust classifier on BIGRoC's performance. In Table 5 we compare the influence of using a non-adversarial classifier (i.e. $\epsilon = 0$) and adversarial classifiers trained with different threat models' sizes on our proposed method, while the rest of the hyperparameters are fixed. As can be seen, using AT classifiers in BIGRoC enhances the results significantly.

Table 5: Illustration of the robust classifier's threat model effect on BIGRoC's performance, measured in FID, using WGAN-GP trained on CIFAR-10 dataset.

| w/o BIGRoC | w/ BIGRoC | | | |
|---|---|---|---|---|
| | $\epsilon_2 = 0$ | $\epsilon_2 = 0.25$ | $\epsilon_2 = 0.5$ | $\epsilon_2 = 1$ |
| 22.32 | 19.71 | 18.79 | 16.28 | 15.87 |

## 7 Discussion and Conclusions

In this work, we propose a novel method that leverages the perceptually aligned gradients phenomenon for refining synthesized images. Due to the core ability of such a robust classifier to better capture significant visual features, it is capable of effectively improving the output of generative models. Our approach does not require additional training of the generative model, and it is completely model agnostic, contrary to other methods. In a line of experiments, we show that our method is highly effective and capable of substantially enhancing the qualitative and quantitative results of various generative models over multiple datasets. Moreover, we conduct an opinion survey that validates that the outputs of BIGRoC are indeed more perceptually pleasing. Interestingly, our work reveals the surprising generative capabilities of robust classifiers, as a single such model can refine the outputs of both simple and SOTA generators.

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

# A    Additional Ablations and Demonstrations

## A.1    The effect of BIGRoC's number of steps

When $\epsilon$ is fixed, increasing the number of steps leads to smaller steps. Fine-grained steps can lead to better performance, but with a computational cost. In Table 6, we summarize the effect of the number of steps w.r.t a fixed $\epsilon$. As can be concluded, 7 is a plausible value since it is a good trade-off between refinement performance and computational cost.

Table 6: The effect of the number of steps in BIGRoC's algorithm, measured in FID, using WGAN-GP trained on CIFAR-10 dataset.

| w/o BIGRoC | w/ BIGRoC | | | |
|---|---|---|---|---|
| | 1 STEP | 7 STEPS | 20 STEPS | 30 STEPS |
| 22.32 | 17.48 | 16.28 | 16.15 | 16.27 |

## A.2    Visual demonstration of BIGRoC's iterations

BIGRoC is an iterative boosting algorithm, and as such, it performs several update steps. In Figure 7, we visualize the optimization algorithm performed by our method. As can be seen, the perceptual quality of the images obtained by BIGRoC gradually improves during its application. More specifically, at first, BIGRoC focuses on the coarse shape and details of the image and then on the fine details.

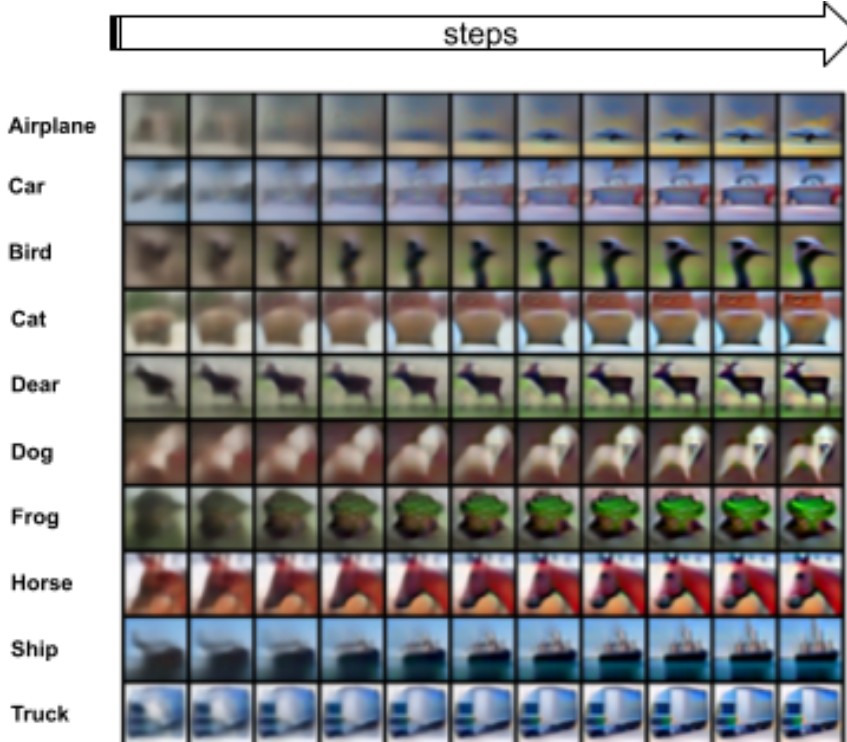

Figure 7: **BIGRoC's steps visualization**, demonstrated using VAE trained on CIFAR-10: On the left column we show the generated images of the model. The righmost column corresponds with the final output of BIGRoC. The middle 9 columns are the results obtained after each intermediate step of our algorithm.

### A.3 Qualitative comparison with non-robust models

As explained in the experimental part of the paper, we study the effects of applying BIGRoC with non-robust classifiers. More specifically, we do so with a non-robust ResNet-50. In Figure 8, we show that applying our algorithm with such a non-robust network does not refine the image as a robust network that has PAG does. We use exactly the same hyperparameters in these experiments.

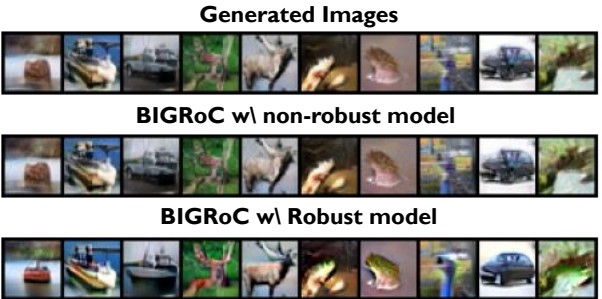

Figure 8: **Guiding classifier effect**. Qualitative comparison between the refinement obtained by BIGRoC with robust and non-robust models guidance.

## B  Opinion Survey Details

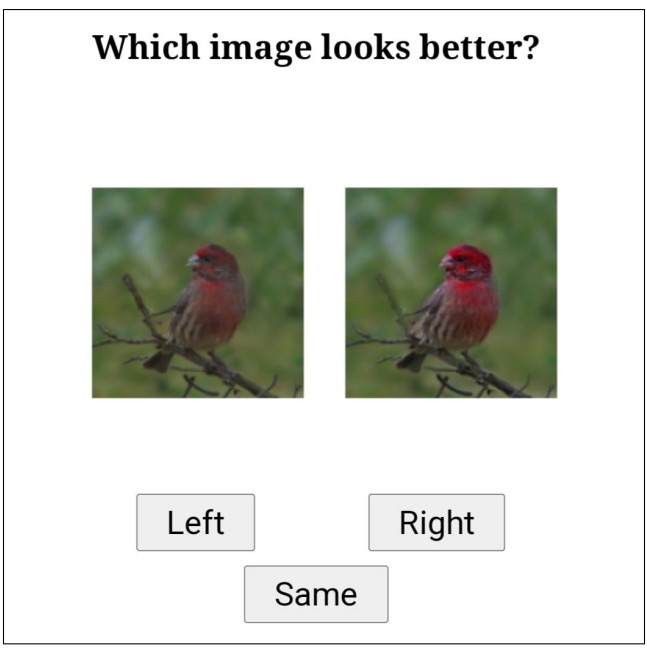

Figure 9: **Opinion survey**. An example of a question from the human-observer survey.

As mentioned in the paper, we conduct an opinion survey in which we ask human evaluators to choose their preferred image from a pair of two – a generated one and BIGRoC's output. In addition, the evaluators were allowed to choose that they have no preference. We randomly generate 100 images using guided-diffusion (Dhariwal & Nichol, 2021) trained on ImageNet $128 \times 128$ and applied BIGRoC on them. We create pairs of original-boosted images in random order and shuffled the order of them for each evaluator. An example of the screen shown to evaluators is displayed in Figure 9. We provide all 100 pairs in the supplementary material.

# C  Implementation Details

In this work, except for three basic generative models (see description in C.1), we did not train models and used only available pretrained ones from verified sources. For quantitative evaluation, we use common implementations of IS and FID metrics. In all of our experiments, we use the same robust classifier to boost all the generative models that operate on the same dataset. We use the pretrained generators to synthesize sets of images and BIGRoC to refine them. The specific details regarding the experimental results are listed in the sections below.

## C.1  CIFAR-10

**Adversarial Robust Classifier** We use a pretrained robust ResNet-50 on CIFAR-10, provided in (Engstrom et al., 2019). This model is adversarially trained with a threat model $\Delta = \{\delta \; : \; \|\delta\|_2 \leq 0.5\}$ with step size $\alpha$ of 0.1.

**Image Generators** Besides VAE, DCGAN, and cGAN that we trained from scratch, using the relevant available codebases, we did not train any other generator and used only publicly available ones. For cGAN-PD, WGAN-GP, SNGAN, InfoMaxGAN, and SSGAN, we use the ones from mimicry repository[1]. We use the pretrained versions of BigGAN and Differential Augmentation CR BigGAN (Diff BigGAN) from the data-efficient GANs repository[2].

**BIGRoC hyperparameters** As stated above, we use the same robust classifier in all our experiments. Thus, the remaining hyperparameters to tune are $\epsilon$, step size, and the number of steps. We empirically find out that the number of steps is relatively marginal (see Section A.1), and thus we opt to use 7. In all the experiments, we fix the step size to be $1.5 * \frac{\epsilon}{num\_steps}$. Since we test various image generators of different qualities, each requires a particular amount of visual enhancement, defined by the value of $\epsilon$. Low-quality generators require substantial improvement and therefore benefit from high $\epsilon$ values, while better ones benefit from smaller values. In the below table, we summarize the value of $\epsilon$ for each tested architecture.

Table 7: BIGRoC's $\epsilon$ values in CIFAR-10 experiments

| ARCHITECTURE | $\epsilon$ | NORMALIZATION |
|---|---|---|
| VAE | 25 | ✓ |
| DCGAN | 5 | ✓ |
| WGAN-GP | 5 | ✓ |
| SNGAN | 3 | ✓ |
| SSGAN | 3 | ✓ |
| cGAN | 5 | ✓ |
| cGAN-PD | 2 | ✓ |
| BIGGAN | 1 | ✓ |
| DIFF BIGGAN | 1 | ✓ |

Where normalization is referred to images in [-1, 1]. To better interpret the meaning of $\epsilon$ in terms of pixels modification, the average change of a pixel value is expressed by $\frac{\epsilon}{\sqrt{32 \times 32 \times 3}}$. For example, in DCGAN, $\epsilon = 5$ is equivalent to an average change of $\approx 0.1$. We note that in this example, the pixels are in a range between -1 to 1 and therefore, the mean change is $\approx 5\%$

## C.2  ImageNet

**Adversarial Robust Classifier** We use a pretrained robust ResNet-50 on ImageNet, provided in (Engstrom et al., 2019), on both $128 \times 128$ and $256 \times 256$. This model is adversarially trained with a threat model $\Delta = \{\delta \; : \; \|\delta\|_2 \leq 3\}$ with step size of 0.5.

---

[1] https://github.com/kwotsin/mimicry
[2] https://github.com/mit-han-lab/data-efficient-gans/tree/master/DiffAugment-biggan-cifar

**Image Generators** We did not train any generator and utilize the publicly available ones. For SNGAN, SSGAN, and InfoMaxGAN, we use the ones from the mimicry repository. As for BigGAN-deep (truncation= 1.0) and guided diffusion, we utilize the fact that BIGRoC can operate on standalone images and utilize the sets of generated images, published in guided diffusion's repository[3], and we apply our method upon these. We test our method using the aforementioned sets of generated images using two setups – with and without the ground truth labels. When operating without the labels, we produce pseudo labels using our robust classifier and then apply BIGRoC.

**BIGRoC hyperparameters** As in CIFAR-10, we only tune the value of $\epsilon$. In the below table, we report the used values for our tested architectures.

Table 8: BIGRoC's $\epsilon$ values in ImageNet experiments

| RESOLUTION | ARCHITECTURE | $\epsilon$ | NORMALIZATION |
|---|---|---|---|
| 128 | SNGAN | 40 | ✓ |
| | SSGAN | 40 | ✓ |
| | INFOMAXGAN | 40 | ✓ |
| | BIGGAN-DEEP | 5 | ✗ |
| | GUIDED DIFFUSION | 1.5 | ✗ |
| 256 | BIGGAN-DEEP | 1 | ✗ |
| | GUIDED DIFFUSION | 1.5 | ✗ |

### C.3  Comparison with other methods

**Adversarial Robust Classifier** For CIFAR-10 we use the same robust classifier as described in Appendix C.1 and for ImageNet we use the same model as in Appendix C.2.

**Image Generators** To fairly compare BIGRoC and the competitive methods, we use BIGRoC to refine the outputs of the same pretrained model as in DOT, DDLS, and DG-$f$low – SN-ResNet-GAN[4]. We experiment in both CIFAR-10 and ImageNet and compare our results with the reported ones of the other methods. The missing values in Table 3 stem from the fact that DOT did not report its results on SN-ResNet-GAN on CIFAR-10, and DG-$f$low was not tested on ImageNet or any other high-resolution dataset.

**BIGRoC hyperparameters** We report in the table below the value of epsilon used to attain the results in Table 3.

Table 9: BIGRoC's $\epsilon$ values in Table 3 experiments

| DATASET | $\epsilon$ | NORMALIZATION |
|---|---|---|
| CIFAR-10 | 1.8 | ✗ |
| IMAGENET | 15 | ✗ |

## D  Fidelity-Diversity Analysis

In image synthesis, fidelity and diversity measure the photorealism and distribution faithfulness of the generated images. In the main paper, we considered FID and Inception Score as our quantitative metrics for evaluating the quality of the generated and boosted images. While IS mainly focuses on fidelity, FID accounts for both. However, to better understand BIGRoC's contribution in terms of fidelity and diversity, we report in Table 10 the precision and recall scores Naeem et al. (2020) of applying our method to different generative models, trained on ImageNet $128 \times 128$. Theoretically, using too large $\epsilon$ values when applying

---

[3]https://github.com/openai/guided-diffusion
[4]https://github.com/pfnet-research/sngan_projection

our method can improve the samples' fidelity but potentially sacrifice the diversity. However, we abstain from this by choosing proper $\epsilon$ values. This way, we improve the overall generation quality, expressed by better FID scores. We demonstrate the effect of $\epsilon$ in Figure 6 – using too large $\epsilon$ values boosts the fidelity (expressed via IS) but harms the diversity (reflected in FID). However, as can be seen from the table, a proper choice of $\epsilon$ enables us to improve the overall quality, expressed by the FID score, without trading off diversity for fidelity.

Table 10: Precision-Recall results on ImageNet $128 \times 128$.

| Architecture | Resolution | Precision ↑ | Recall ↑ | FID ↓ |
|---|---|---|---|---|
| SNGAN (Miyato et al., 2018) | $128 \times 128$ | 0.25 | 0.43 | 62.28 |
| w/ BIGRoC$_{\text{PL}}$ | | 0.32 | 0.52 | 40.40 |
| Δ | | **+0.07** | **+0.09** | **-21.88** |
| SSGAN (Chen et al., 2019) | $128 \times 128$ | 0.27 | 0.43 | 63.60 |
| w/ BIGRoC$_{\text{PL}}$ | | 0.33 | 0.51 | 38.93 |
| Δ | | **+0.06** | **+0.08** | **-24.67** |
| InfoMaxGAN (Lee et al., 2021) | $128 \times 128$ | 0.27 | 0.46 | 60.61 |
| w/ BIGRoC$_{\text{PL}}$ | | 0.33 | 0.52 | 37.70 |
| Δ | | **+0.06** | **+0.06** | **-22.91** |
| ADM (Dhariwal & Nichol, 2021) | $128 \times 128$ | 0.70 | 0.65 | 5.91 |
| w/ BIGRoC$_{\text{GT}}$ | | 0.72 | 0.64 | 4.41 |
| Δ | | **+0.02** | **-0.01** | **-1.5** |
| Guided Diffusion (Dhariwal & Nichol, 2021) | $128 \times 128$ | 0.78 | 0.59 | 2.97 |
| w/ BIGRoC$_{\text{GT}}$ | | 0.79 | 0.59 | 2.75 |
| Δ | | **+0.01** | **-** | **-0.22** |

## E    Qualitative Results

In this section, we show additional qualitative results to further demonstrate the qualitative enhancement attained by our method. We use image generators of different qualities, both conditional and unconditional, and show the generated images and the boosted ones in Figures 10, 11, 12 and 13. The images below are simply the 100 first synthesized images from each class.

## F    Debiasing

In Section 3 we describe our debiasing algorithm, which aims to induce uniform class distribution over the outputs of BIGRoC. The rationale behind this procedure is expressed in Equation 2. To better understand the outcome of our debiasing, we present in Figure 14 its effect visually. We compare the results of applying BIGRoC in the unconditional case, with and without debiasing. One can clearly see that it reduces the amount of the majority class and leads to a more uniform class distribution, by modifying the images accordingly.

## G    Computational Resources

In this work, we do not train generative models (when possible) and use publicly available pretrained ones to generate images and apply BIGRoC on their outputs. This way, we decouple the generative model's complexity from our algorithm since it is applied to the generated images themselves. In all of our experiments, we use the Google COLAB service with a single GPU.

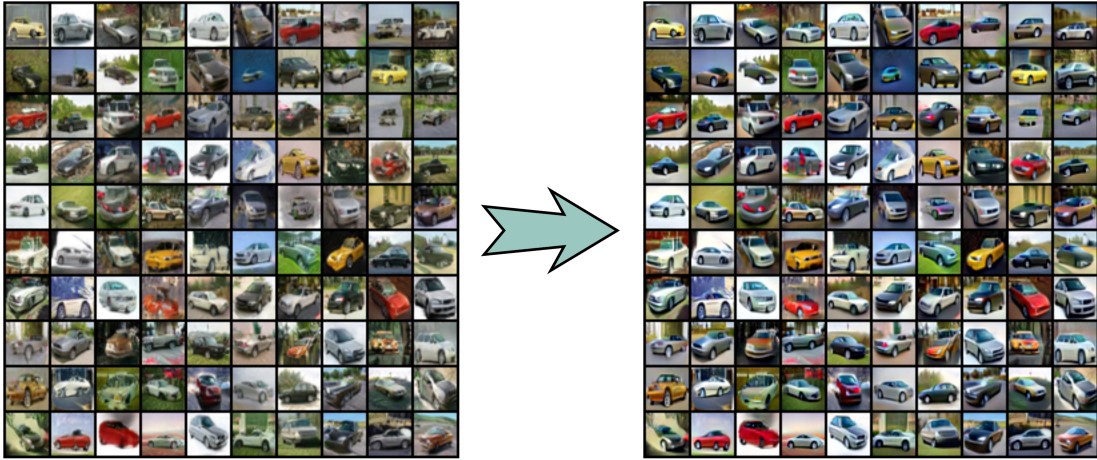

(a) A comparison between BigGAN generated images of class automobile and the corresponding boosted ones.

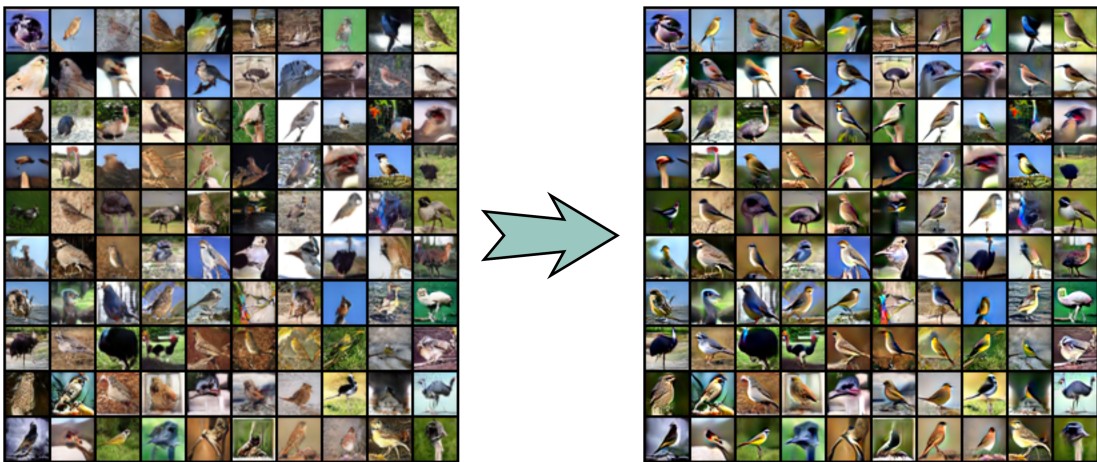

(b) A comparison between BigGAN generated images of class bird and the corresponding boosted ones.

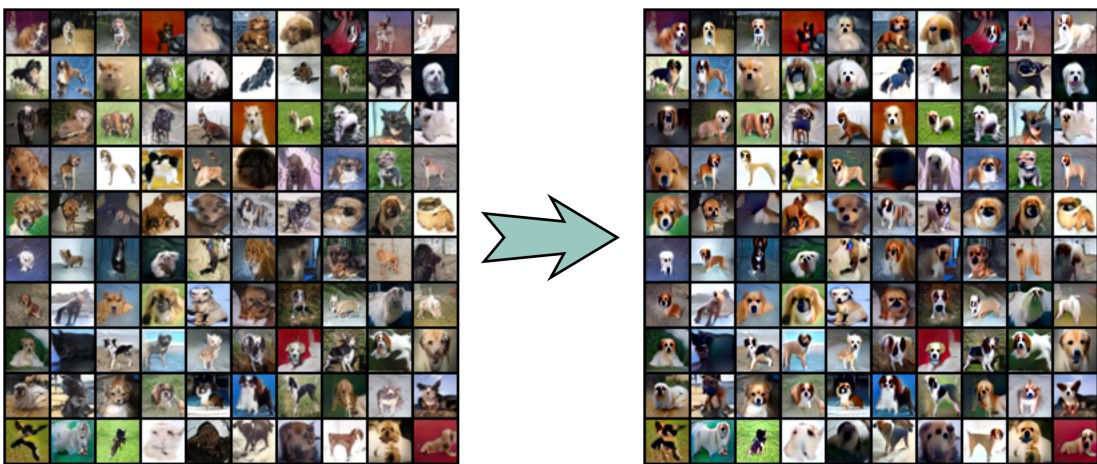

(c) A comparison between BigGAN generated images of class dog and the corresponding boosted ones.

Figure 10: **BIGRoC refinement of BigGAN**. A qualitative comparison between BigGAN generated images of CIFAR-10 samples and the refined ones by BIGRoC algorithm.

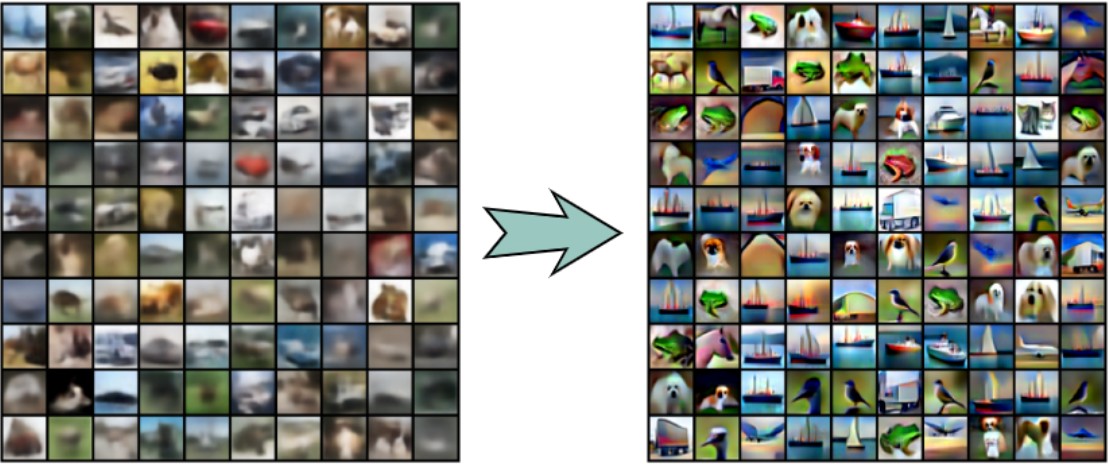

Figure 11: **BIGRoC refinement of VAE**. A qualitative comparison between an unconditional VAE generated images of CIFAR-10 samples and the refined ones by BIGRoC algorithm.

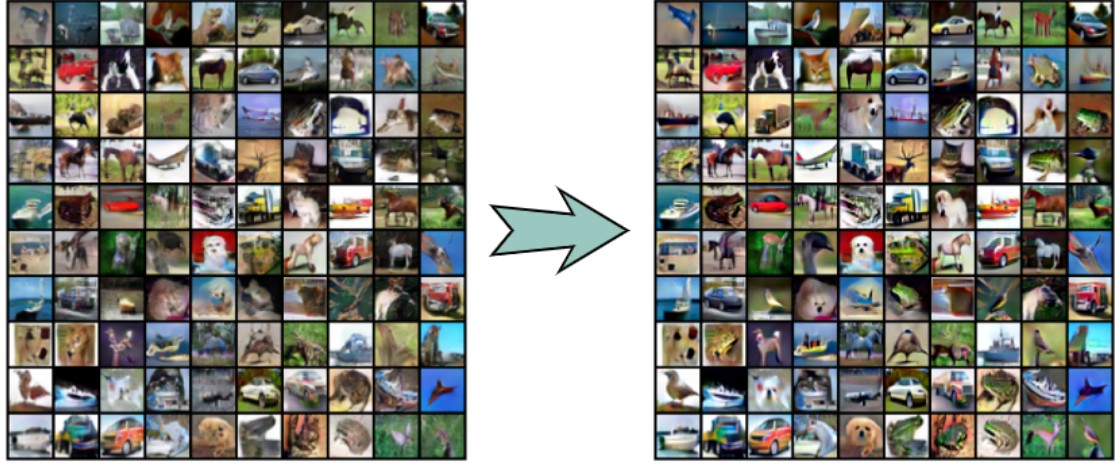

Figure 12: **BIGRoC refinement of WGAN-GP**. A qualitative comparison between an unconditional WGAN-GP generated images of CIFAR-10 samples and refined ones by BIGRoC algorithm.

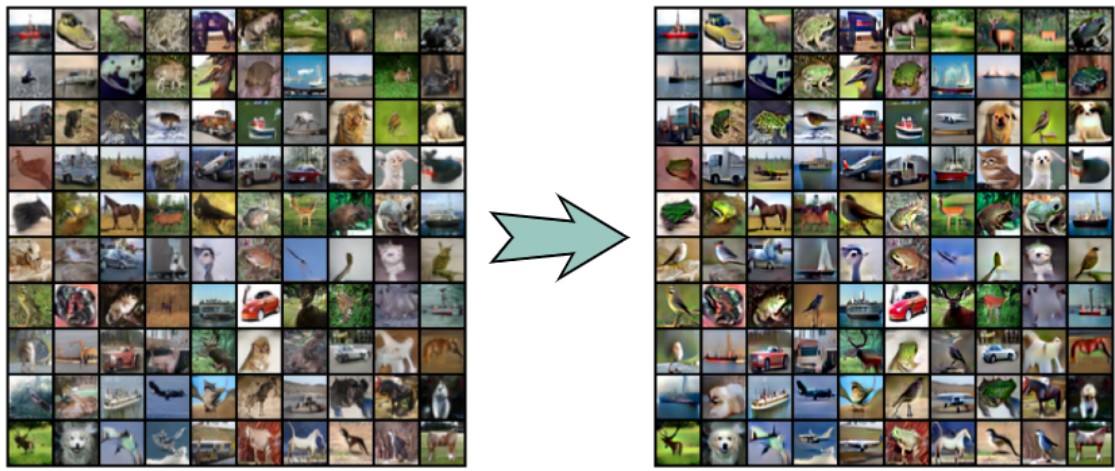

Figure 13: **BIGRoC refinement of SSGAN**. A qualitative comparison between an unconditional SSGAN generated images of CIFAR-10 samples and refined ones by BIGRoC algorithm..

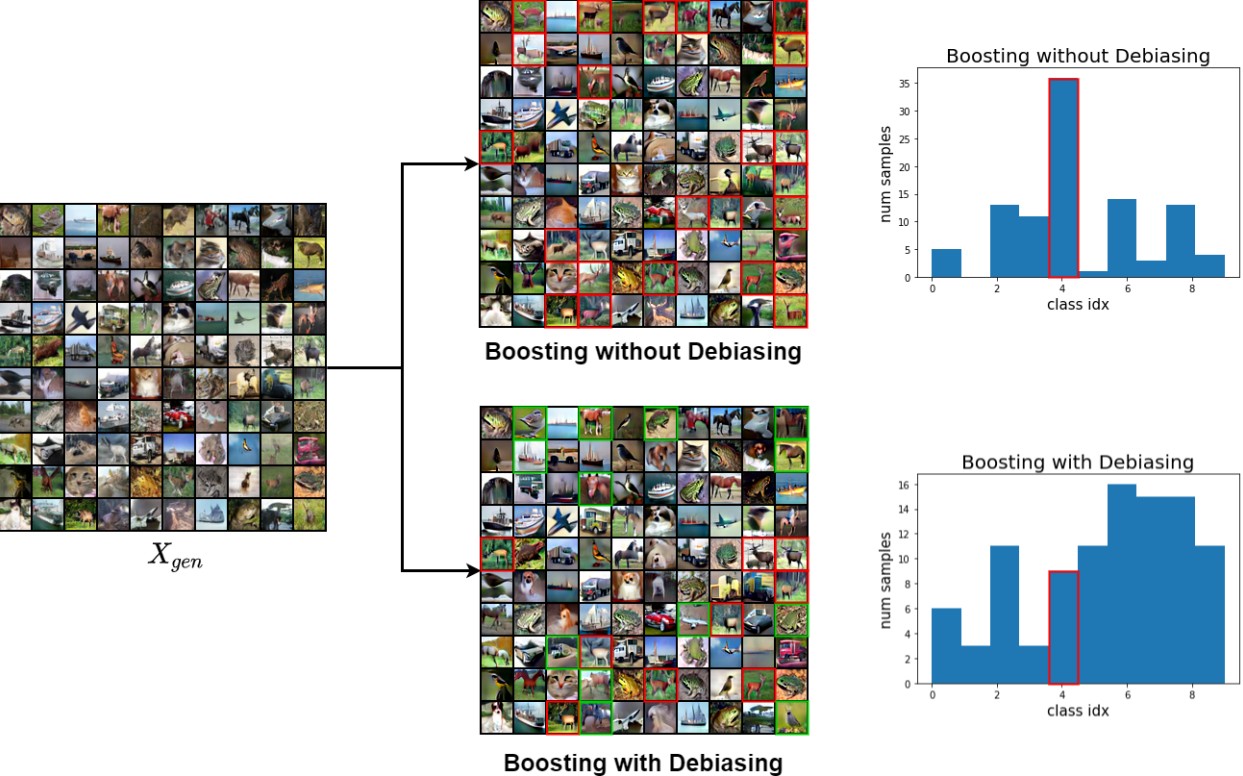

Figure 14: **Demonstration of the debiasing technique**. We show 100 generated images by an unconditional SNGAN and the results of the BIGRoC algorithm, with and without the proposed debiasing. As can be seen, the outputs of the boosting algorithms are perceptually superior, while the histograms expose the fact that the suggested debiasing algorithm induces a more uniform labels distribution. In the "Boosting without Debiasing" experiment, 36 out of 100 images are classified as deers, and only 3 are horses. The most prominent deer images are marked in red. However, when applying the debiased boosting, the number of deers is reduced to 9, and the number of horses is increased to 15. We mark the boosted images that remain deer in red, and images that are modified to other minority classes in green. As can be seen, many of the deers were changed to be horses, a perceptually similar class.

