# OpenReview forum: "BIGRoC: Boosting Image Generation via a Robust Classifier"
_TMLR — Accepted by TMLR_

### Review · Reviewer_5TM6 · 2022-12-16

**Summary Of Contributions:**

The paper proposes a method named BIGRoC, which performs image refinement by running PGD w.r.t. an adversarially trained (a.k.a. robust) classifier.  The authors further propose ways to class-balance the generation when not given GT class labels.  In terms of FID and IS, BIGRoC showed improved results on CIFAR and ImageNet datasets.

**Audience:**

Yes

**Claims And Evidence:**

Yes

**Requested Changes:**

A major question that remains unanswered for me is whether the improved image quality comes with a cost of diversity.  Intuitively, increasing class-conditional likelihood will make the image look more specific to the object class, and appear more visually appealing. But from a generative modelling perspective, distribution coverage and sample diversity are equally important.  Both IS and FID are pretty simplistic metrics that do not capture image diversity well and the human preference study was done as an image-level where the human subject cannot judge distribution-level performance.  Here are some suggestions of how this question might be answered:
1. In terms of evaluation, how does it do in terms of data likelihood?
2. It’s intuitive that with class-dependent gradient the image quality improves due to moving towards a local optimum in the likelihood space, but does that come at a sacrifice of “coverage”?
What about class-dependent FID, and other non-parametric metrics like those suggested in “Reliable Fidelity and Diversity Metrics for Generative Models”?

The comparison to baseline methods in image refinement also seem to be very brief (only Table 3).

While I do think this paper studies an interesting problem, and the proposed method seems reasonable and shows some good results, I think this paper can benefit the readers more if the above question is answered.


**Strengths And Weaknesses:**

## Strength
* Paper is clearly written
* Shows improved image quality on benchmark datasets.

## Weaknesses
* There isn’t enough results (either theoretical or empirical) for validating and understanding the proposed method. See Requested Changes below.
* The impact of this study can be limited as current SOTA image generation models based on large foundation models are substantially better than the models used in the paper.

---

> ### Author Response · Authors · 2022-12-28
> **Authors reply to reviewer 5TM6**
>
> Thank you for your valuable input and constructive feedback. We address below the changes requests and weaknesses.
>
> **Fidelity and Diversity** - Indeed, fidelity and diversity are both essential for image generation. Theoretically, using too large $ \epsilon $ values when applying our method can improve the samples' fidelity but potentially sacrifice the diversity. However, we abstain from this by choosing proper $\epsilon$ values. This way, we improve the overall generation quality, expressed by better FID scores. We demonstrate the effect of $\epsilon$ in Figure 6 -- using too large $\epsilon$ values boosts the fidelity (expressed via IS) but harms the diversity (reflected in FID).
>
> As FID is a single metric that accounts for both fidelity and diversity, it does not imply whether better scores stem from the former, the latter, or both. Thanks to your suggestion, we study the effects of BIGRoC in terms of diversity and fidelity using the precision and recall metrics. Specifically, we apply our method on several generative models trained on ImageNet $128\times128$ and report the obtained results below:
>
> | Method | Precision | Recall| FID|
> |-----------|---------------|---------|-----|
> |SNGAN| 0.25 | 0.43|62.28|
> | +BIGRoC| 0.32 | 0.52|40.40|
> |SSGAN | 0.27 | 0.43 |63.60|
> | +BIGRoC| 0.33 | 0.51|38.93|
> |InfoMaxGAN | 0.27 | 0.46 | 60.61|
> | +BIGRoC| 0.33 | 0.52| 37.70 |
> |ADM| 0.70 | 0.65| 5.91|
> |+BIGRoC|0.72 | 0.64 | 4.41|
> |ADM-G| 0.78 | 0.59| 2.97|
> |+BIGRoC|0.79 | 0.59 | 2.75|
>
> The results on GANs indicate that our method is capable of improving both fidelity and diversity. When applying BIGRoC to an unguided diffusion model (ADM) that is known for its mode coverage (reflected in the recall metric), our method is not capable of improving the diversity but instead trades off diversity for fidelity in a way that obtains better overall generation quality (expressed via FID scores). We note that a similar trend occurs in [1], where better generation was obtained by trading off diffusion diversity for fidelity. Applying BIGRoC to ADM-G improves the overall generation quality by boosting fidelity without sacrificing diversity.
>
> We thank the reviewer for this important note and will include these results in the revised version.
>
> **Comparison to baseline methods** - We compare our approach with state-of-the-art image refinement algorithms and adopt the same settings in the respective papers to ensure a fair comparison. We experiment with SN-ResNet-GAN as it is the one used in the baseline methods used for comparison. To make our comparison more extensive, we conduct experiments on both CIFAR-10 and ImageNet $128\times128$. Moreover, besides the quantitative experiments, we conceptually compare the differences between our and existing methods to highlight our contributions and advantages over the baseline methods.
>
> **Study impact and relevancy** - This paper considers various generative models of different qualities and empirically demonstrates BIGRoC refinement capabilities on the most common datasets (CIFAR and ImageNet). As for our top-performing model, we used ADM-G, which is still amongst the best methods for image synthesis of ImageNet, according to the existing benchmarks. As stated correctly, the image generation field evolves rapidly, and currently, there are image synthesizers of very high quality (e.g., stable diffusion). However, these models are mainly text-to-image models and do not report quantitative results on datasets such as ImageNet. In fact, the evaluation used in such works stems from qualitative comparison and user studies rather than quantitative metrics such as FID, IS, precision, and recall. These factors make it difficult to properly assess the improvements of an image refinement method on such models. However, we agree that extending our approach to text-to-image
>
> [1] Diffusion Models Beat GANs on Image Synthesis - Dhariwal and Nicol

---

### Review · Reviewer_AYWs · 2022-12-19

**Summary Of Contributions:**

This paper propose an new strategy to enhance the quality of synthesized images by introducing a robust classifier. The classifier can capture the semantic information, which covers the shortage of a generative model. It is interesting that a classifier for high-level vision task can help a low-level vision task. The authors conduct extensive experiments on ImageNet and CIFAR, where images with high quality are generated.

**Audience:**

Yes

**Broader Impact Concerns:**

No concerns.

**Claims And Evidence:**

Yes

**Requested Changes:**

See the weakness.

**Strengths And Weaknesses:**

Strength：

-This paper uses a robust classifier to enhance the ability of generative models. It is glad to see that this method can be combined with generative models, includings WGAN, SNGAN, SSGAN, etc.

- The authors conduct extensive ablation study to investigate the effectiveness of each components.

Weakness:

-The classifier improve image's quality by discovering adversarial examples. There are many methods about adversarial examples, imposing the classifier can be designed with different formulation. It is interesting to investigate whether the formulation of classifier affect the quality of  generated images.

-The robust classifier seems a general method. Besides image synthesis, could the method be used for other low-level tasks, such as Image super-resolution？

---

> ### Author Response · Authors · 2022-12-25
> **Authors reply to reviewer AYWs**
>
> We thank the reviewer for the valuable input and address below the raised issues.
>
> - We agree that the choice of the robust classifier and its formulation is important to examine. In Section 6.3, we study the effect of $\epsilon$ when training a robust classifier using an $L_2$-based threat model. Thanks to your suggestion, we conduct an experiment investigating the effects of applying BIGRoC with a robust classifier trained with an $L_\infty$ threat model. Specifically, we utilize a pretrained ResNet-50 trained against adversarial examples using $L_\infty$ ball with $\epsilon=4/255$ on ImageNet $128\times128$ and apply BIGRoC to ADM-G generated images. We keep the rest of the hyperparameters as in BIGRoC with $L_2$ based robust model. We compare the quantitative results with the generated images and the refined ones using an $L_2$ based robust classifier and report the results below. As the results suggest, while both types of robust classifiers successfully refine the generated images, the model trained with the $L_2$ threat model leads to better results. We will include these results in our revised version upon acceptance.
>
> | Method | FID | IS |
> |-----------|------|-----|
> |ADM-G|2.97| 141.37|
> |BIGROC $L_2$| 2.53 | 169.73|
> |BIGRoC $L_\infty$| 2.93 | 161.12 |
>
> - Perceptually Aligned Gradients are a trait of robust classifiers that can be used for generative tasks and can be harnessed for both image generation and sample refinement but also tasks such as super-resolution. For example, for super-resolution, given a robust classifier on the higher resolution, one can naively upsample the subsampled image and modify it to maximize the conditional probability estimated by the robust classifier.

---

### Review · Reviewer_Jnwa · 2022-12-20

**Summary Of Contributions:**

This paper presents a model-agnostic framework to boost existing visual generation models with a robust classifier. Only post-processing can improve the visual quality without any model finetuning. This is inspired by the adversarial robustness that the small perturbation would not change the prediction. The authors assume that modifying an image to sharpen such a classifier’s decision yields visual features perceptually aligned with the target class. Thus, this paper introduces the BIGRoC, which can be applied for both conditional and unconditional generation given an adversarially trained classifier. BIGRoC has shown large potential on a variety of benchmarks, including ImageNet and CIFAR.




**Audience:**

Yes

**Claims And Evidence:**

Yes

**Requested Changes:**

N/A

**Strengths And Weaknesses:**

Pros:

+ The motivation of this paper is clear and strong. Applying a robust classifier to boost visual generation is straightforward but meaningful.

+ The whole paper is well written, with clear logic to follow. The discussion of related works is comprehensive and includes most of the related literature.

+ The proposed model is efficient and is supposed to be computationally friendly. There is no finetuning of generation models, which usually takes much effort.

+ I have not noticed a similar previous work to apply a robust classifier to boost the visual generation model. The overall idea is novel.


Cons:

- The refinements in visual quality seem marginal (I have not seen much difference between the two methods in Fig.1). And some FID score also shows a trivial improvement.

- There are many hyper-parameters to be assigned. This may need some hand-craft tricks to make it work, which is not friendly for new developers.

- Some important experiments are missing. I recommend the authors establish a user study for comparison, which could be an important evaluation matrix since the authors claimed that the robust classifier could help generative results better align with human perception.

Question:

Will authors have any plans to extend this framework to text-to-image generation?

Is it any discussion of weakness?

What is the time/computation cost of your proposed post-processing?

---

> ### Author Response · Authors · 2022-12-25
> **Authors reply to reviewer Jnwa**
>
> We thank the reviewer for the valuable input and address below the raised issues.
>
> **Cons**:
>
> - The level of the required visual refinement depends on the quality of the generated image, as high-quality generated images do not require substantial modification. It affects the level of improvement in the quantitative and qualitative evaluation when applying our method to different generative models. Thus, in Figure 1, where we apply BIGRoC to a top-performing model (ADM-G), the norm of the modifications is relatively low. Nevertheless, the BIGRoC yields better images (sharper with more realistic colors). It is also supported by our human evaluation survey described in Section 5.3. However, when we apply our method to lower-quality image generators, it yields significant refinement, both quantitatively and qualitatively (please see Figures 3 and 11 for example).
>
> - Indeed, according to Algorithm 2 describing BIGRoC, there are some hyperparameters –  $\epsilon$, step size, and the number of steps (T). In Appendix A1, we study the effect of the number of steps. We find that $T=7$ is sufficient for refinement and use it for all our experiments. As for the step size, we use a rule of thumb according to which we set it to be $1.5 * \epsilon / T$, making it a non-tunable hyperparameter (as described in Appendix C1). Thus, the only hyperparameter that needs to be tuned is $\epsilon$ which defines the level of the required refinement. Nevertheless, one can use the heuristic that better generative models require a smaller change for tuning $\epsilon$.
>
> - We completely agree with the importance of conducting a user study and established one (described in Section 5.3). The results of this survey strongly attest that BIGRoC results are preferred by human evaluators.
>
> **Questions**:
>
> - We are exploring the fascinating direction of extending this work to text-to-image generation schemes.
>
> - In this paper, we focus on sample refinement of generative models trained on closed academic datasets (CIFAR-10 and ImageNet). Recent advancements in vision-language research led to text-to-image generators capable of generating multi-object scenes. Moreover, such models can generate general content images that are not represented in regular datasets. Thus, BIGRoC which is based on a single-object multi-class classifier, cannot be applied to refine such models.
>
> - The application of BIGRoC is similar to conducting a 7-step targeted white-box adversarial attack and therefore depends on the choice of the robust classifier. For example, applying BIGRoC using a ResNet50 on a $128\times128$ image takes 0.16 seconds using a single Tesla T4 GPU. We will add a computation cost analysis to our revised version upon acceptance.

---

> > ### Comment · Reviewer_Jnwa · 2023-01-12
> > **Post Rebuttal**
> >
> > Dear Authors,
> >
> > Thanks for the detailed response. Most of my concerns are addressed. The remaining minor weakness is the user study. There are only 100 image pairs to compare, which seems to be pretty tiny. And the discussion of statistical significance should be included. I am leaning toward accepting the submission and hope the authors to further improve the manuscript in the final version.
> >
> > Best,
> > Reviewer

---

> > > ### Author Response · Authors · 2023-01-14
> > > **Post Rebuttal**
> > >
> > > Dear reviewer,
> > >
> > > We sincerely thank you for the discussion and constructive feedback. We will improve the manuscript based on the feedback from the reviewers for the final version.
> > >
> > > Thanks,
> > > The authors.

---

### Decision · Action_Editors · 2023-01-22

**Recommendation:** Accept as is

**Comment:**

This paper proposes a model-agnostic method to improve the quality of synthesized images. The proposed robust classifier is reasonable, whose effectiveness is verified by extensive experiments. All the reviewers recognize the contribution of this paper.

During the rebuttal phase, the authors post more explanation (e.g., generalization of the proposed method) and experimental analysis (e.g., experiments with several generative models trained on ImageNet 128x128 ). These responses answer most of the questions proposed by reviewers. All the reviewers post positive recommendations (i.e., 'Leaning Accept' and 'Accept'). Therefore, I recommend acceptance of this paper.

**Audience:**

Yes.

**Claims And Evidence:**

Yes. The proposed method is reasonable. The authors also conduct extensive experiments to investigate its effectiveness.